cellular biology

metabolomics, polycyclic aromatic hydrocarbons, toxicity, biomarker, oxidative stress

**Author for correspondence:**
Jian Yang
e-mail: jiany@ffrc.cn

# Effects of 9,10-phenanthrenequione on antioxidant indices and metabolite profiles in *Takifugu obscurus* plasma

## Shulun Jiang[1], Jian Yang[1] and Dian Fang[2]

[1]Wuxi Fisheries College, Nanjing Agricultural University, Wuxi 214081, People's Republic of China
[2]Freshwater Fisheries Research Center, Chinese Academy of Fishery Sciences, Wuxi 214081, People's Republic of China

(iD) SJ, 0000-0002-5863-8010

Derived from polycyclic aromatic hydrocarbons (PAHs), oxygenated-PAHs (oxy-PAHs) may pose hazards to aquatic organisms, which remain largely unknown. *Takifugu obscurus* is an important anadromous fish species of high economic and ecological values. In the present study, *T. obscurus* was acutely exposed to $44.29\ \mu g\,l^{-1}$ 9,10-phenanthrenequione (9,10-PQ) for 96 h. Changes of antioxidant indices and metabolite profiles in plasma were compared between 9,10-PQ treatment and the control. The results showed that 9,10-PQ treatment significantly increased malondialdehyde (MDA) content during 6 to 96 h, increased superoxide dismutase (SOD) and catalase (CAT) activities at 6 h, but decreased them at 96 h. These results indicated that 9,10-PQ induced oxidative stress to fish. Ultra-performance liquid chromatography-mass spectrometry (UPLC-MS) analysis revealed that four metabolic pathways were influenced in response to treatment with 9,10-PQ, including glycerophospholipid metabolism, phenylalanine, tyrosine and tryptophan biosynthesis, purine metabolism and sulfur metabolism. These pathways are associated with antioxidant mechanisms, biosynthesis of neurotransmitters and innate immune functions. Thus, the as-obtained results confirmed that 9,10-PQ induced oxidative stress and raised concerns of neurotoxicity and immunotoxicity to fish. Overall, the present study posed a high environmental risk of oxy-PAHs to aquatic ecosystems.

# 1. Introduction

Polycyclic aromatic hydrocarbons (PAHs) are a group of typical persistent organic pollutants widespread in natural environments and have drawn great attention due to their mutagenic and carcinogenic toxicity [1,2]. This family of compounds consists of two or more fused benzene rings and contain derivatives with alkyl, nitrogen or oxygen substitutions [3]. Among them, 16 PAHs have been stipulated as the prior pollutants by the Environmental Protection Agency of the USA (US EPA). PAHs are produced during incomplete combustion of organic materials and fossil fuels. More importantly, accidental oil spill, leakages from drilling operations and industrial wastewater release plenty of PAHs to aquatic environments [4]. Following the Gulf War oil spill event in 1991, PAHs could be detected in local seawater even after three years, with the concentration of total PAHs ranging from 21.14 to 320.5 µg l$^{-1}$ [5]. During the oil spill of the Deepwater Horizon accident, severe PAH pollution was observed and the concentration of total PAHs reached 151 mg l$^{-1}$ [6]. Such high levels of PAHs would undoubtedly affect aquatic organisms.

During the past decades, most studies have focused on the toxicity of parent-PAHs to fish, displaying increased deformities, lesions, tumours and other toxic effects [7]. After exposure to PAHs mixture, zebrafish embryos exhibited morbid syndromes, including cardiac dysfunction, oedema, spinal curvature and abnormal development of craniofacial structures [8]. Cultivation with PAH-contaminated sediments resulted in hepatic lesions in mummichogs (*Fundulus heteroclitus*) [9]. Prevalence of liver tumour in brown bullheads (*Ameiurus nebulosus*) was even used as an indicator of PAHs contamination [10]. During the emission of total PAHs, oxygenated-PAHs (oxy-PAHs) may also be released, which are derivatives of parent-PAHs. Moreover, after releasing in the natural environments, parent-PAHs can be transformed to oxy-PAHs [11]. However, toxicity and environmental risks of oxy-PAHs to fish have not been well investigated.

The pufferfish *Takifugu obscurus* is a commercially important species [12], since it is a traditional and popular food in Asian countries. However, the natural resource of this species has been declining rapidly owing to overexploitation and environmental pollution [13]. As an anadromous fish species, *T. obscurus* may swim across various freshwater and seawater environments, thus exhibiting a high risk of being exposed to PAHs [14], since PAHs have been broadly detected in aquatic environments. To the best of our knowledge, only Wang [15] compared the hepatic toxicity of three PAHs to *T. obscurus*. The results showed that treatments with 400 µg l$^{-1}$ phenanthrene, 400 µg l$^{-1}$ 3-methyl phenanthrene and 100 µg l$^{-1}$ 9,10-phenanthrenequinone significantly increased glutathione transferase (GST) activity. Treatments with 100 µg l$^{-1}$ phenanthrene, 100 µg l$^{-1}$ 3-methyl phenanthrene and 12.5 µg l$^{-1}$ 9,10-phenanthrenequinone significantly elevated malondialdehyde (MDA) contents. These results preliminarily revealed that PAHs induced significant oxidative stress to *T. obscurus*. More investigations are still required to comprehensively understand toxicity of PAHs and their derivatives to *T. obscurus*, which may contribute to the protection of natural resource and food safety of *T. obscurus*.

Activities of superoxide dismutase (SOD), catalase (CAT) and content of MDA are traditional oxidative/antioxidant indices to evaluate harmful effects of xenobiotics on organisms [16,17]. Treatments with the water-soluble fraction of Arabian crude oil which mainly contained PAHs mixture, significantly ($p < 0.05$) lowered SOD activity in the European sea bass *Dicentrarchus labrax* [18] and increased MDA content in *Lateolabrax japonicus* livers [19]. At the transcriptional level, numbers of researches have been implemented to explore oxidative stress induced by PAHs in zebrafish and other aquatic organisms [20–23]. However, the effects of oxy-PAHs on changes of these antioxidant indices in *T. obscurus* have not been reported. Endogenous metabolites are the most proximal indicators, which can reflect metabolic alterations in response to exogenous stimuli [24,25]. The metabolomics technology, which detects the changes of all metabolites in organisms, has been widely applied to investigate metabolic alterations in response to xenobiotics [26]. Among the methods used for metabolomics, liquid chromatography–mass spectrometry (LC-MS) is the most compatible technique for detection of small metabolites and displays robust reliability and reproducibility [27,28]. The LC-MS-based metabolomics has been successfully used to explore effects of two PAHs (benz[a]anthracene and benz[a]anthracene-7,12-dione) on metabolisms of zebrafish, demonstrating toxic influences on protein biosynthesis, mitochondrial function, neural development, vascular development and cardiac function [29]. Thus, metabolomics analyses might be useful to explore influences of oxy-PAHs on fish at the metabolic level.

9,10-phenanthrenequione (9,10-PQ) is an oxygenated derivate of the typical PAH phenanthrene. Plasma is a pool of metabolites and can systematically reflect metabolic changes in the whole body [30]. Thus, plasma is a suitable subject to investigate physiological changes of organisms in response to pollutants. In the present study, to assess the environmental risk of 9,10-PQ, *T. obscurus* juveniles were exposed to 9,10-PQ and changes of antioxidant indices and metabolite profiles in plasma were investigated. These results may provide basic information to understand the environmental effects of PAHs on aquatic ecosystems.

# 2. Material and methods

## 2.1. Animals

Juvenile *T. obscurus* ($24.96 \pm 0.97$ g, $10.19 \pm 0.45$ cm) were obtained from the Freshwater Fisheries Research Centre, Chinese Academy of Fishery Science (Wuxi, China). Before exposure, fish were acclimated in glass aquaria containing 50 l of dechlorinated tap water for two weeks. The water was aerated constantly. The pH, temperature and dissolved oxygen content (DO) of water were controlled at $7.8 \pm 0.05$, $30 \pm 1°C$ and $6.9 \pm 0.1$ mg l$^{-1}$, respectively. Fish were fed twice a day under a photoperiod of 14 h : 10 h (light : dark). During this period, no fish died.

## 2.2. Exposure and sampling

9,10-PQ (purity > 98%) was purchased from ChemService (Worms, Germany). A stock solution of 44.29 mg l$^{-1}$ 9,10-PQ was prepared in dimethyl sulfoxide (DMSO) and stored at 4°C in dark. The previous acute toxicity test in our laboratory discovered that the 96 h-LC$_{50}$ of 9,10-PQ to *T. obscurus* was approximately 88.59 µg l$^{-1}$. In the present study, the exposure concentration was set at 50% 96 h-LC$_{50}$ (44.29 µg l$^{-1}$) as high exposure concentration to test the acute toxicity of 9,10-PQ, which could maximize the acute toxic effect as well as not cause death. About 1‰ DMSO was used as the control. Fish were randomly distributed into treatment and control in three biological replicates. Each replicate contained 50 fish. The exposure lasted for 96 h and 75% of the culture media were renewed daily.

During the exposure period, three fish were randomly sampled from each aquarium after 0, 2, 6, 12, 24, 48 and 96 h to determine antioxidant indices. After 96 h, four fish were randomly sampled from each aquarium for metabolome analysis. Approximately 0.5 ml of blood was collected from the caudal vein and immediately transferred into ethylenediaminetetraacetic acid (EDTA) centrifuge tube. After centrifugation at 5250$g$ at 4°C for 10 min, plasmas were snap-frozen in liquid nitrogen and preserved at −80°C.

## 2.3. Determination of oxidative stress biomarkers

Contents of MDA, SOD and CAT activities were measured following the methods of Placer *et al.* [31], Sun *et al.* [32] and Beer & Sizer [33], respectively. One unit of SOD activity was defined as that enzyme quantity which causes 50% suppression of nitroblue tetrazolium (NBT) reduction. One unit of CAT activity was defined as the activity required to destroy 1 nmol of H$_2$O$_2$ at 25°C in 0.2 M phosphate buffer for 1 min.

## 2.4. Metabolomics analysis

For each sample, equal amount of plasmas from four fish in the same replicate were pooled. Then, 100 µl of pooled plasmas were mixed with 300 µl of methanol to precipitate proteins, and 10 µl of 2.9 mg ml$^{-1}$ DL-O-chlorophenylalanine was added as the internal standard. The mixture was vortexed for 30 s and centrifuged at 18 000$g$ and 4°C for 15 min. Finally, 200 µl of supernatant was transferred to new vials. After evaporation of solvent, the residues were dissolved in 100 µl of 5% acetonitrile solution containing 0.1% formic acid.

Four microlitres of samples were injected into an UPLC-QTOF/MS system (Agilent 1290 Infinity LC System, Agilent 6530 UHD and Accurate-Mass Q-TOF, Agilent Technologies, Santa Clara, CA, USA) equipped with an electrospray ionization (ESI) source and a C18 column ($2.1 \times 100$ mm, 1.8 µm, Agilent Technologies). The mobile phase was 0.1% formic acid in water (mixture A), and 0.1% formic acid in acetonitrile (mixture B). The flow rate was 0.35 ml min$^{-1}$. The column temperature was 40°C and the automatic injector temperature was 4°C. The samples were eluted following the procedure: 0%–5% at 1 min, 5%–20% from 1 to 6 min, 20%–50% from 6 to 9 min, 50%–95% from 9 to 13 min, and 95% from 13 to 15 min.

Mass spectrometry was performed at both positive (ESI$^+$) and negative ion modes (ESI$^-$), separately. Source temperature was 100°C and gas flow rate was 50 l h$^{-1}$. The desolvation temperature was 350°C and the flow rate was 600 l h$^{-1}$. The capillary voltage and cone voltage were 4 and 35 kV in positive ion mode, respectively, while 3.5 and 50 kV in negative ion mode, respectively. Data from 50 to 1000 $m/z$ were collected with the scan time of 0.03 s and the interval time of 0.02 s. Leu-enkephalin was used as the lock mass ([M$^+$H]$^+$ of 556.2771 Da in ESI$^+$ and [M$^-$H]$^-$ of 554.2615 Da in ESI$^-$). The

pooled sample was used as quality control (QC) to validate the reproducibility and stability of the method and equipment.

## 2.5. Data processing and pattern recognition

Raw UPLC-MS data was first processed using Mass Profiler software (Agilent), including peak detection, peak alignment and peak integration. Subsequently, peak areas of all samples were normalized to the total area. Afterwards, a two-dimensional data matrix was generated, including retention time (RT), compound molecular weight (mass) and peak intensity. The normalized data were subjected to SIMCA-P 14.1 (Umetrics AB, Umea, Sweden) for orthogonal partial least-squares discriminant analysis (OPLS-DA) to compare metabolic differences between treatment and control. Important variables on the projection (VIP) of the OPLS-DA model were used to discover the potential variables contributing to the differentiation. Differentially expressed metabolites (DEM) were defined as VIP > 1 and $p < 0.05$. The METLIN web-based metabolomics database (https://metlin.scripps.edu) was used for tentative identification of significant features, including accurate mass and MS/MS spectral matching [34]. Verification of the identified metabolites was conducted using an in-house library of standards based on the accurate mass and retention time of metabolite standards provided by IROA Technologies.

## 2.6. Analysis of metabolic pathways

Pathway analyses were performed based on the metabolome of *Danio rerio* and the Kyoto Encyclopedia of Gene and Genomes (KEGG) database [35] using the MetaboAnalyst 2.0 [36]. The KEGG enrichment analyses were statistically conducted using the Fisher's exact test.

## 2.7. Statistical analysis

The variation (upregulation and downregulation) of DEMs were expressed as $\log_2$(fold change). Antioxidant indices are presented as mean ± standard deviation (s.d.). Student's *t*-test was applied to compare significant differences between treatment and control. $p < 0.05$ was considered statistically significant.

# 3. Results

## 3.1. Changes of oxidative stress biomarkers

Along with exposure time, SOD, CAT activities and MDA content did not significantly change in the control. However, in 9,10-PQ treatment, SOD and CAT activities increased first but then decreased, MDA contents increased at early time but then remained constant. Compared with the control, treatment with 9,10-PQ significantly increased MDA content at 72.97%, 91.21%, 85.84%, 93.37% and 97.44% respectively from 6 h to 96 h ($p < 0.05$); SOD and CAT activities in 9,10-PQ treatment were significantly higher than those in the control at 6 h with the increase of 45.18% and 44.54%, respectively, but significantly lower than those in the control at 96 h with the decrease of 29.86% and 32.62% respectively ($p < 0.05$, figures 1–3).

## 3.2. Total ion current chromatograms

The total ion current (TIC) chromatograms of QC samples tested before and after experiments showed reproducible patterns in both ESI$^+$ and ESI$^-$ models (figure 4$a$,$b$), demonstrating high stability and repeatability of UPLC-MS analyses. The TIC chromatograms of plasma samples differed between treatment and control in either ESI$^+$ or ESI$^-$ models. For instance, the peak of the control at 3.6 min disappeared in 9,10-PQ treatment in the ESI$^+$ model. The differences in TIC chromatograms indicated that treatment with 9,10-PQ influenced the metabolite profiles in fish plasma (figure 4).

## 3.3. Orthogonal partial least-squares discriminant analysis analyses and differentially expressed metabolites

OPLS-DA analysis clearly discriminated treatment with 9,10-PQ and the control in plots (figure 5). All samples were located within the 95% confident intervals (black circle in figure 5). R$^2$Y values indicate the contribution rate and $Q^2$ values represent the predictive ability of the supervision model. In ESI$^+$ model,

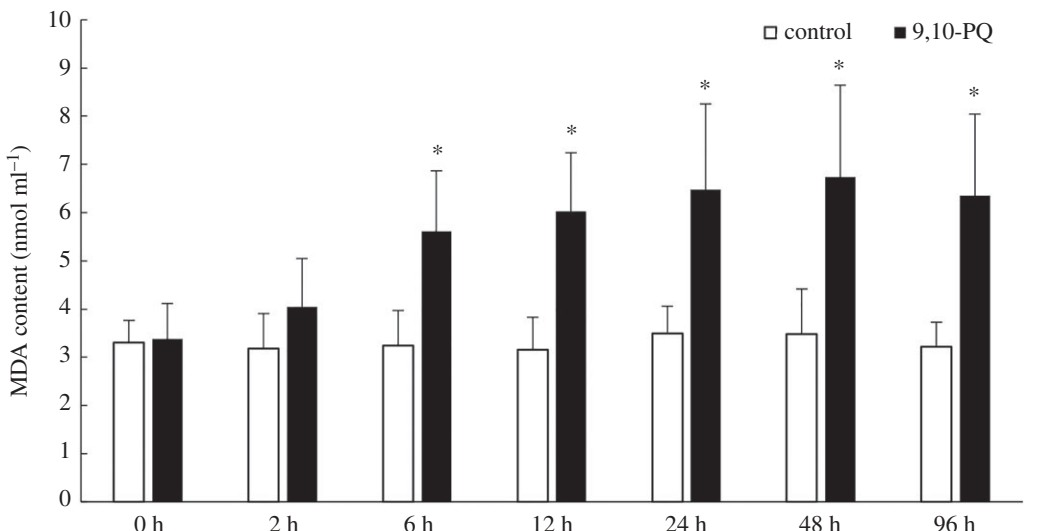

**Figure 1.** MDA content in plasma of *T. obscurus* under sublethal exposure to 9,10-PQ. The data indicate mean (nine fish samples from three aquaria) and s.d. Significant differences ($p < 0.05$) between control and 9,10-PQ treatment are marked by asterisks (Student's *t*-test).

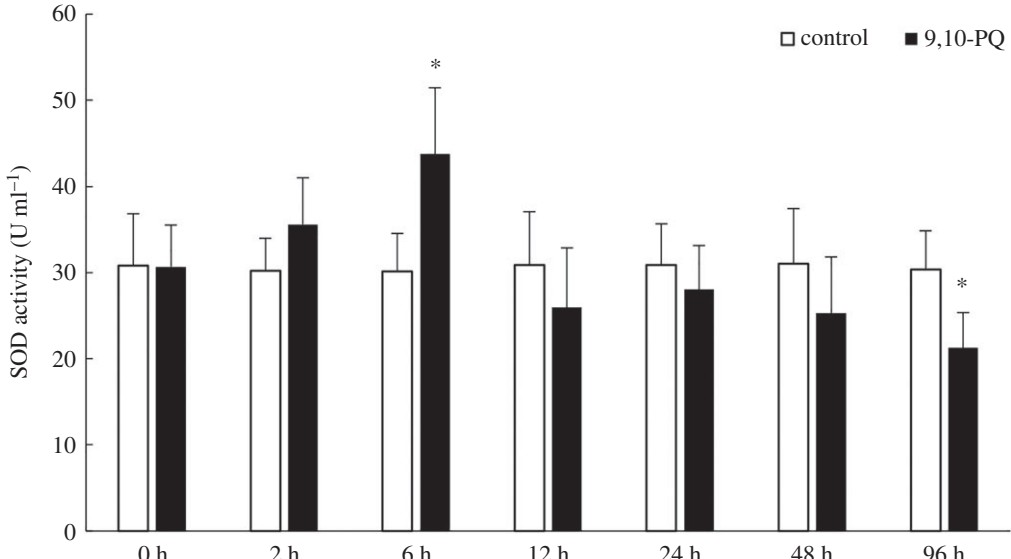

**Figure 2.** SOD activity in plasma of *T. obscurus* under sublethal exposure to 9,10-PQ. The data indicate mean (nine fish samples from three aquaria) and s.d. Significant differences ($p < 0.05$) between control and 9,10-PQ treatment are marked by asterisks (Student's *t*-test).

$R^2Y$ and $Q^2$ were 0.988 and 0.625, respectively; in ESI$^-$ model, $R^2Y$ and $Q^2$ were 0.949 and 0.601, respectively. The high $R^2Y$ and $Q^2$ values in the present study indicated reliability of the OPLS-DA models.

Based on the OPLS-DA results, DEM were identified. Overall, 21 metabolites were differentially expressed between 9,10-PQ treatment and the control in ESI$^+$ model and 10 DEMs in ESI$^-$ model. Compared with the control, LPA(0:0/16:0), L-tryptophan, dehydrojuvabione, PS(15:1(9Z)/22:6(4Z,7Z,10Z,13Z,16Z,19Z)) and cinnamaldehyde were upregulated greatly with $\log_2$(fold change) higher than 2; N$\alpha$-acetyl-L-glutamine and guanosine were downregulated remarkably with $\log_2$(fold change) lower than −2 (table 1).

## 3.4. Enrichment of Kyoto Encyclopedia of Gene and Genomes pathways

In the ESI$^+$ model, two pathways, glycerophospholipid metabolism, and phenylalanine, tyrosine and tryptophan biosynthesis were significantly enriched. In the ESI$^-$ model, purine metabolism and sulfur

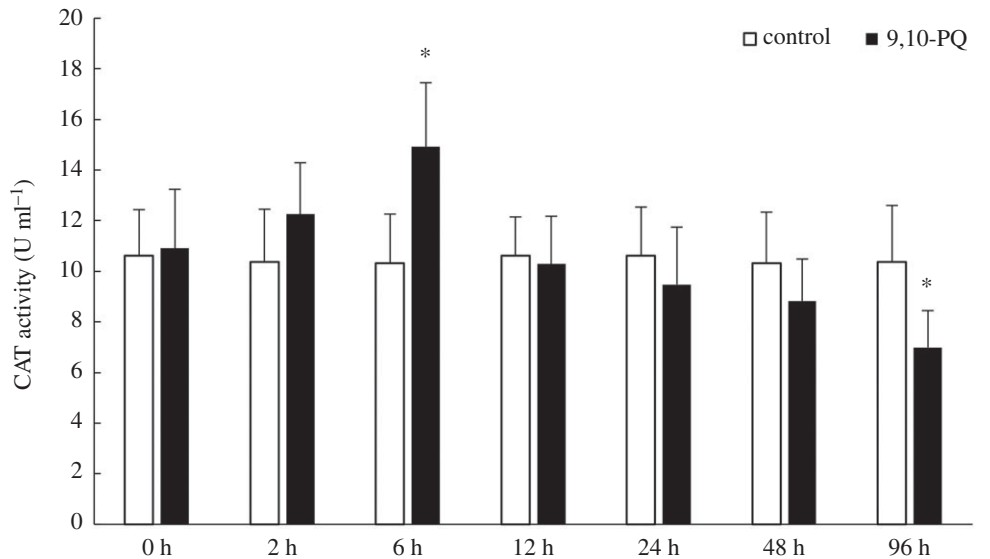

**Figure 3.** CAT activity in plasma of *T. obscurus* under sublethal exposure to 9,10-PQ. The data indicate mean (nine fish samples from three aquaria) and s.d. Significant differences ($p < 0.05$) between control and 9,10-PQ treatment are marked by asterisks (Student's *t*-test).

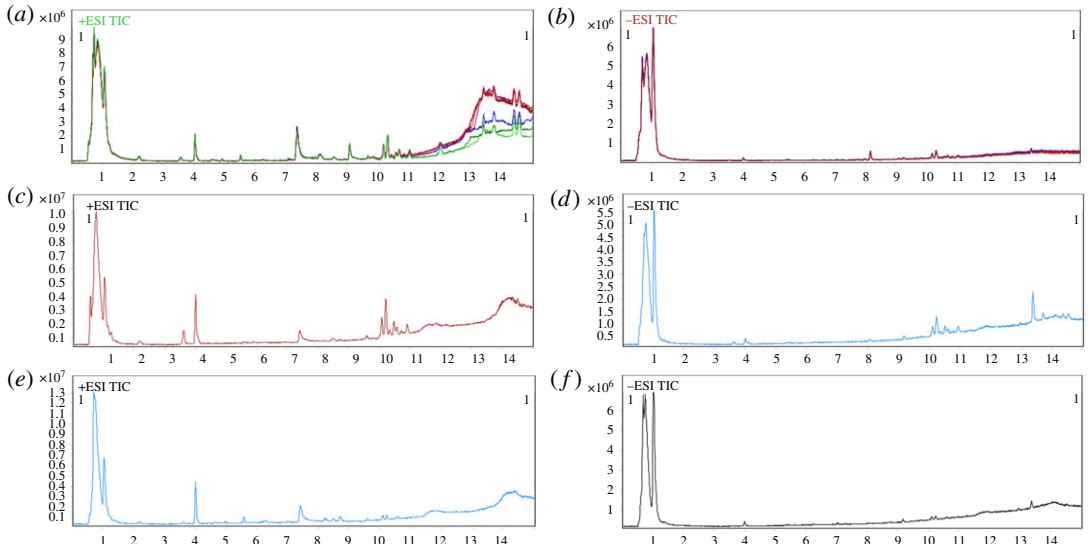

**Figure 4.** Typical UPLC-MS TIC chromatograms of (*a*) QC in positive ion mode; (*b*) QC in negative ion mode; (*c*) the control sample in positive ion mode; (*c*) the control sample in negative ion mode; (*e*) 9,10-PQ sample in positive ion mode; (*f*) 9,10-PQ sample in negative ion mode.

metabolism were significantly enriched. The purine metabolism involved three DEMs (inosine, guanosine and sulfuric acid) and four lysophosphatidylcholines (LysoPCs) contributed to the glycerophospholipid metabolism (figure 6 and table 2).

## 4. Discussion

Phenanthrene is one of the 16 prior PAH pollutants stipulated by US EPA, and its toxicity to aquatic organisms has been well studied. However, its oxygenated derivative 9,10-PQ has been seldom investigated. *T. obscurus* is an important migratory fish species and reveals a high risk of exposure to fossil oil, which contains high content of PAHs and 9,10-PQ. The high-throughput metabolomics can reveal metabolic alterations, which can directly reflect the toxic effects of exogenous pollutants. The present study applied the UPLC-MS-based metabolomics technology to investigate plasma responses

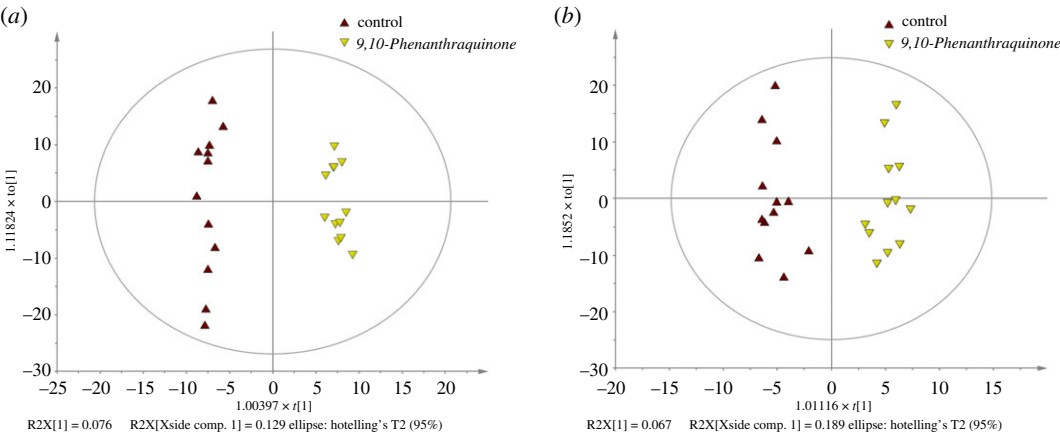

**Figure 5.** OPLS-DA score plot of metabolic profiles in *T. obscurus* plasma. (*a*) Positive ion mode; (*b*) negative ion mode. Brown triangles indicate the plasma samples of the control while yellow triangles indicate the plasma samples of 9,10-PQ group.

to acute 9,10-PQ exposure in *T. obscurus*. The results showed that significant oxidative stress was induced and a total of four KEGG pathways were significantly influenced, which represented the major toxic effects of 9,10-PQ on physiological metabolism in *T. obscurus*.

MDA content represents the level of lipid peroxidation (LPO). SOD and CAT are two typical antioxidant enzymes. These three indices are frequently used to assess oxidative stress induced by xenobiotics [37]. In the present study, after exposure to 9,10-PQ, MDA content significantly increased, suggesting peroxidation of cell membranes. Similarly, Lin *et al.* [19] reported that treatment with pyrene increased MDA contents in *Lateolabrax japonicus*. Exposure to $200\,\mu g\,l^{-1}$ phenanthrene significantly increased the LPO values in the muscle of estuarine guppy *Poecilia vivipara* at the level of Bayes factor > 24 [38]. Treatment with benzo(a)pyrene increased MDA contents in *Ruditapes philippinarum* significantly [39]. To resist damages of oxidative stress, antioxidant mechanisms, such as SOD and CAT might be activated [40]. As the most important and powerful antioxidant enzymes, SOD could detoxify superoxide anions while CAT could reduce $H_2O_2$ [39]. In the present study, both SOD and CAT activities significantly increased after exposure to 9,10-PQ for 6 h, which might be responses to oxidative stress. However, after 96 h, their activities were significantly lower than those in the control, probably because over-production of superoxide anions and $H_2O_2$ might exceed the elimination capacity of SOD and CAT, and then damage the antioxidant defence system. This result was in accordance with previous research which discovered the same pattern of SOD and CAT activities in *Carassius auratus* in response to phenanthrene exposure [41]. These results demonstrated that oxy-PAHs exhibited oxidative effects on fish, which was similar to parent-PAHs.

Glycerophospholipid metabolism was significantly ($-\log(p) = 3.880$) affected in response to 9,10-PQ exposure. Similarly, this pathway was also altered by phenanthrene in human keratinocytes HaCaT cells [42]. Glycerophospholipids are the main lipids in cell membrane [43,44]. They regulate membrane potential, curvature and influence membrane fusion, fission and ion transport [45]. Thus, altered glycerophospholipid metabolism might then influence cellular functions. LysoPCs are major lipid components of plasma derived from phosphatidylcholines (PC) by the enzyme lecithin-cholesterol acyltransferase (LACT) [46,47]. LysoPCs play important roles in regulating immunologic cellular functions, peculiarly in priming neutrophils, which are main cells of innate immunity [48]. In the present study, four LysoPCs were involved in this pathway. Among them, three LysoPCs were downregulated and one LysoPC was upregulated in response to 9,10-PQ treatment. The overall downregulation of LysoPCs in the present study might negatively affect structure and functions of cellular membranes as well as the innate immunity.

As previously reported, treatments with PAHs (benz[a]anthracene and benz[a]anthracene-7,12-dione) exhibited neurobehavioral toxicity to zebrafish (*Danio rerio*) by affecting the phenylalanine, tyrosine and tryptophan biosynthesis pathway and downregulating dopamine, serotonin, phenylalanine, tyrosine and tryptophan [29]. Similarly, developmental neurotoxicity was observed in Japanese medaka embryos when exposed to benz[a]anthracene [49]. Moreover, He *et al*. [50] demonstrated that exposure to pyrene led to neurodevelopmental lesions in *Sebastiscus marmoratus* embryo. In the present study, neurotransmitters were detected, probably because these compounds are mainly localized to the neural tissues rather than plasma. However, treatment with 9,10-PQ also significantly affected the phenylalanine, tyrosine and tryptophan biosynthesis pathway ($-\log(p) = 3.426$), which contained two upregulated DEMs

**Table 1.** Differentially expressed metabolites derived from OPLS-DA in treatment with 9,10-PQ compared with the control. FC: fold change.

| VIP | name | detected mass | RT (min) | *p*-value (*t*-test) | log$_2$(FC) |
|---|---|---|---|---|---|
| ESI$^+$ | | | | | |
| 2.417 | dodecanedioic acid | 230.152 | 10.226 | 0.000 | 0.932 |
| 2.410 | guanosine | 283.0918 | 1.397 | 0.000 | −1.433 |
| 2.386 | LPA(0:0/16:0) | 410.2288 | 4.034 | 0.001 | 4.527 |
| 2.229 | coenzyme Q4 | 454.3268 | 9.834 | 0.002 | 1.385 |
| 2.203 | L-tryptophan | 204.0788 | 11.990 | 0.002 | 2.621 |
| 2.065 | spermidine | 145.158 | 0.565 | 0.004 | 1.557 |
| 2.051 | LysoPC(20:4(5Z,8Z,11Z,14Z)) | 543.3323 | 10.188 | 0.004 | −1.029 |
| 1.900 | L-phenylalanine | 165.0791 | 2.174 | 0.009 | 0.778 |
| 1.899 | dehydrojuvabione | 264.1733 | 11.427 | 0.010 | 2.123 |
| 1.893 | malic acid | 134.0229 | 1.394 | 0.010 | −1.846 |
| 1.888 | ectoine | 142.0742 | 0.820 | 0.010 | 1.208 |
| 1.814 | PS(15:1(9Z)/22:6(4Z,7Z,10Z,13Z,16Z,19Z)) | 791.4415 | 4.931 | 0.014 | 2.839 |
| 1.770 | dodecylphosphocholine | 351.2411 | 11.950 | 0.017 | 1.059 |
| 1.716 | LysoPC(22:5(4Z,7Z,10Z,13Z,16Z)) | 569.348 | 10.755 | 0.021 | 1.372 |
| 1.594 | homoglutamine | 160.0887 | 11.658 | 0.034 | 0.257 |
| 1.592 | LysoPC(22:6(4Z,7Z,10Z,13Z,16Z,19Z)) | 567.3328 | 10.175 | 0.034 | −0.907 |
| 1.556 | octylamine | 129.1516 | 4.293 | 0.039 | −1.423 |
| 1.548 | LysoPC(15:0) | 481.3166 | 11.598 | 0.040 | −0.589 |
| 1.544 | PC(13:0/0:0) | 453.2854 | 10.360 | 0.040 | −0.775 |
| 1.532 | propionyl-L-carnitine | 217.132 | 1.410 | 0.042 | 1.035 |
| 1.501 | cinnamaldehyde | 132.0625 | 0.915 | 0.047 | 2.159 |
| ESI$^-$ | | | | | |
| 2.143 | 2-acetolactic acid | 132.0437 | 38.989 | 0.007 | 1.337 |
| 2.009 | Nα-acetyl-L-glutamine | 188.0799 | 50.817 | 0.013 | −3.370 |
| 2.008 | LysoPC(15:0) | 481.3167 | 697.672 | 0.013 | −0.925 |
| 1.928 | glutathione, oxidized | 612.1526 | 47.434 | 0.017 | 1.370 |
| 1.924 | guanosine | 283.0918 | 83.546 | 0.018 | −2.062 |
| 1.848 | 3-furoic acid | 112.0166 | 46.445 | 0.023 | 0.756 |
| 1.777 | dehydrochorismic acid | 224.0357 | 516.608 | 0.030 | 0.442 |
| 1.760 | inosine | 267.0273 | 239.943 | 0.032 | 0.615 |
| 1.744 | dehydroascorbic acid | 174.0166 | 71.069 | 0.034 | 1.245 |
| 1.715 | sulfuric acid | 97.9677 | 50.209 | 0.037 | 1.016 |

(L-phenylalanine and L-tryptophan). L-phenylalanine is an essential amino acid to animals and is also the precursor of tyrosine, catecholamine neurotransmitters, dopamine, norepinephrine and epinephrine [51,52]. These results were not conflicted with the above publications. Inhibition of the transformation process from L-phenylalanine to neurotransmitters would accumulate more L-phenylalanine and L-tryptophan. More investigations are required to validate this hypothesis.

In response to 9,10-PQ exposure, purine metabolism was the most affected pathway (−log(*p*) = 6.843), involving upregulation of inosine and sulfuric acid as well as downregulation of guanosine. Inosine is formed by the deamination of adenosine [53], displaying ability in reduction of oxidative stress and elevation of peroxidase activity [54]. Besides, inosine can also improve immune response, for instance, bactericidal activity and lysozyme activity in Japanese flounder *P. olivaceus* [55]. In the present study,

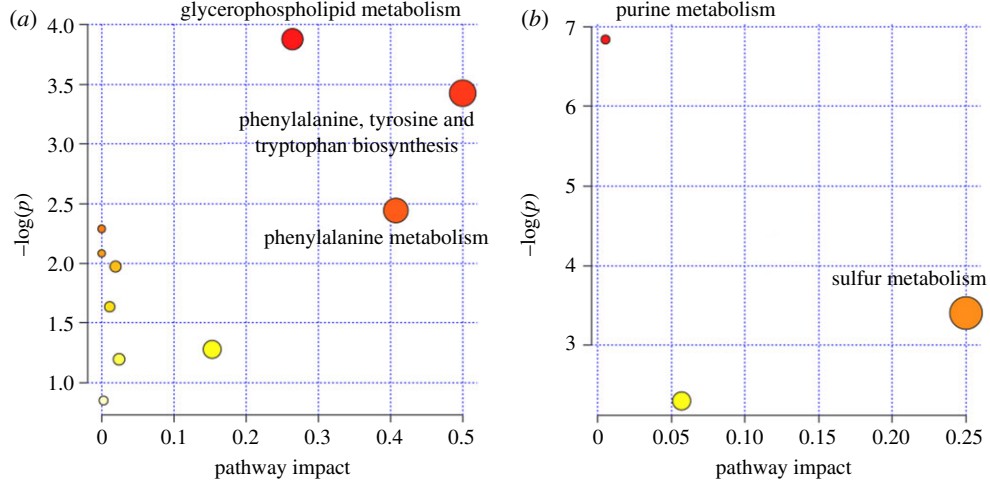

**Figure 6.** Enrichment of metabolic pathways based on differentially expressed metabolites. (*a*) Positive ion mode; (*b*) negative ion mode. All matched pathways are depicted based on −log(*p*) value from pathway enrichment analysis and pathway impact score from pathway topology analysis. Colour gradient indicates the significance of the pathway ranked by −log(*p*) value (red: higher −log(*p*) values and yellow: lower −log(*p*) values). Circle size indicates pathway impact score (the larger the circle the higher the influence score).

**Table 2.** Enrichment of KEGG pathways. DEM: differentially expressed metabolites.

| pathway Name | DEM | −log(*p*) | impact |
|---|---|---|---|
| ESI⁺ | | | |
| glycerophospholipid metabolism | LysoPC(22:6(4Z,7Z,10Z,13Z,16Z,19Z)) | 3.880 | 0.264 |
| | LysoPC(20:4(5Z,8Z,11Z,14Z)) | | |
| | LysoPC(22:5(4Z,7Z,10Z,13Z,16Z)) | | |
| | LysoPC(15:0) | | |
| phenylalanine, tyrosine and tryptophan biosynthesis | L-phenylalanine | 3.426 | 0.500 |
| | L-tryptophan | | |
| ESI⁻ | | | |
| purine metabolism | inosine | 6.843 | 0.005 |
| | guanosine | | |
| | sulfuric acid | | |
| sulfur metabolism | sulfuric acid | 3.405 | 0.25 |

content of inosine increased 1.53-fold in treatment with 9,10-PQ, compared with the control, indicating that *T. obscurus* may produce more inosine to resist oxidative stress and harmful effects on immunity caused by 9,10-PQ. Downregulation of guanosine in the present study was similar to a previous cell culture experiment, in which exposure to 160 μg l⁻¹ PAHs mixture significantly downregulated the level of guanosine due to inhibition of hypoxanthine guanine phosphoribosyl transferase (HGPRT) [56,57]. Oxidation of guanosine is a biomarker of oxidative stress [58]. Considering the increased MDA level in the present study, decreased guanosine content might be a response to oxidative stress, which would negatively affect the repair of DNA damage [57] and the non-specific immunity system [59].

Sulfuric acid is an important metabolite in both purine metabolism and sulfur metabolism pathways. It is an essential component for biosynthesis of antioxidant products, such as glutathione. Both previous publications and the present study demonstrated that oxy-PAHs induced production of reactive oxygen species (ROS) [60]. Thus, the increased content of sulfuric acid in the 9,10-PQ treatment was probably a response of activation of antioxidant status. However, the effect of oxy-PAHs on sulfur metabolism has been poorly studied and further researches are still required. The natural resource of *T. obscurus* has

decreased dramatically in the past few decades. Migration across various habitats increased the risk of being exposed to industrial wastewater, oil spill and leakage. Oxy-PAHs have higher toxicity and water solubility compared with parent-PAHs [61], but have not received international supervision [62]. The present study for the first time investigated effects of 9,10-PQ on plasma metabolites and antioxidant indices in *T. obscurus*. The results displayed that 9,10-PQ induced oxidative stress, neurotoxicity and harmed innate immunity, raising environmental concerns of oxy-PAHs on aquatic ecosystem.

# 5. Conclusion

Acute exposure to 9,10-PQ induced oxidative stress and influenced phenylalanine, tyrosine and tryptophan biosynthesis, glycerophospholipid, purine and sulfur metabolisms in *T. obscurus* plasma. More investigations on the toxicity of oxy-PAHs to *T. obscurus* are required to profoundly understand the hazards of oxy-PAHs.

Ethics. The protocol used in the present study was approved by the Animal Ethics Committee in the Freshwater Fisheries Research Centre of the Chinese Academy of Fishery Sciences. All experimental operations were conducted in accordance with the approved protocol.

Data accessibility. The datasets supporting the results presented in this article are uploaded and available online at the Dryad Digital Repository: https://dx.doi.org/10.5061/dryad.vx0k6djmv [63].

Authors' contributions. All authors made substantial contributions to this paper. S.J. contributed to the experimental design and conducted the experiments. D.F. carried out the statistical analyses. S.J. drafted the manuscript. J.Y. revised the manuscript. All authors approved the final version for publication.

Competing interests. The authors have no competing interests.

Funding. This study was financially supported by the National Infrastructure of Fishery Germplasm Resources (2017DKA3047-003) and the Provincial Key Laboratory of Conservation and Utilization of Important Biological Resources in Anhui.

Acknowledgements. We appreciate Mr. Jun Qiang for his valuable comments on this manuscript.

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
