## [Reviewer comments · Royal Society Open Science]

Review History

RSOS-191356.R0 (Original submission)

Review form: Reviewer 1

Is the manuscript scientifically sound in its present form?

Yes

Are the interpretations and conclusions justified by the results?

Yes

Is the language acceptable?

Yes

Do you have any ethical concerns with this paper?

No

Have you any concerns about statistical analyses in this paper?

No

Recommendation?

Major revision is needed (please make suggestions in comments)

Comments to the Author(s)

The research with title: Effects of 9,10-phenanthrenequinone on antioxidant indices and metabolite profiles in *Takifugu obscurus* plasma, was revised. The article focuses on studying the effect of 9,10-PQ in *T. obscurus*, as well as identify the metabolic pathways was influenced in response, allowing important information to be provided to clarify the effects of these pollutants on aquatic organisms. However, some points need to be improved and clarified in order to be able to consider publishing the work.

In the Introduction section:

1. On line 52, the authors mention the interest in assessing the effects of these pollutants on aquatic organisms. However, in reviewing references 7 and 8, it was found that these articles focus primarily on the effects on human populations, so it is advisable to look for other references focused on aquatic organisms.
2. Check line 53 to 55, the information presented does not match the original reference, since in this study they focused on studying *A. brevis*. In a part of the article they mention that the effects presented by this species is equivalent to the evaluation criteria of bioassays where *Rhepoxinius abronius* is used.

In the material and methods section:

1. On line 53 of page 2, justify the reason for using 50% of the LC50 of 9,10-PQ.
2. In line 55 to 56 on page 2 the authors mention that during the exhibition, the 75% of the culture media were renewed daily. The stability kinetic of 9,10-PQ was performed previously, what then is the actual concentration to which the organisms were exposed? Clarify this point.
3. Is the term "cultural media" correct? check the grammar
4. It is important that authors review the title of point 3.4 "Determination of antioxidant indices". Whereas MDA content levels are a biomarker of oxidative damage or index of damage. In contrast, enzyme activity (SOD and CAT) refers to enzymatic antioxidant status. Check the review the definition of antioxidant indices.

In the results section:

1. In section 4.1, it is recommended to indicate the percentages of the increase or decrease with respect to the control in the times where a significant difference occurred, this in order to appreciate the magnitude of the change in the biomarkers of oxidative stress.

In the discussion section:

1. In line 50 on page 5, specify that the levels of LPO that increased in *P. vivípara* exposed to 200 ug/L phenanthrene were in the muscle.
2. Review the citation on line 59, page 5, [46] the paragraph does not correspond to that reference.
3. Review the numbering of the citations from page 6.

In the reference section:

1. Review reference 41, missing names of authors
2. Review the numbering of references, starting with reference 45

Review form: Reviewer 2

Is the manuscript scientifically sound in its present form?

Yes

Are the interpretations and conclusions justified by the results?

Yes

Is the language acceptable?

Yes

Do you have any ethical concerns with this paper?

No

Have you any concerns about statistical analyses in this paper?

No

Recommendation?

Accept with minor revision (please list in comments)

Comments to the Author(s)

Please include the life stage of the fish species used. This is because a differential response might be observed across male and female or life stage.

Review form: Reviewer 3

Is the manuscript scientifically sound in its present form?

Yes

Are the interpretations and conclusions justified by the results?

Yes

Is the language acceptable?

Yes

Do you have any ethical concerns with this paper?

No

Have you any concerns about statistical analyses in this paper?

Yes

Recommendation?

Accept with minor revision (please list in comments)

Comments to the Author(s)

Manuscript title: Effects of 9,10-phenanthrenequinone on antioxidant indices and metabolite profiles in Takifugu obscurus plasma

Manuscript ID: RSOS-191356

Takifugu obscurus is an interesting fish species due to its unique place in genomics and also due to its value in culinary industry. Authors have presented a comprehensive data on this species under exposure to 9,10-phenanthrenequinone. Manuscript is written in a good manner. However, there are few comments.

1. Selection on one PAH 9,10-phenanthrenequinone needs to be justified. Is this the dominant PAH? What about BaP that is benzo-a-pyrene?
2. Are plasma data sufficient? I mean, some vital tissue such as liver which is critical for metabolism should have been taken.

3. Some data show plateau such as MDA. Then how this response can be correlated with long-term exposure.
4. MDA (I guess lipid peroxidation) is described as an antioxidant (see 3.4 and 4.1).
5. In Figures 1,2 and 3 authors write that significant differences ($p < .05$) between control and 9,10-PQ treatment are marked by asterisks (Student's t-test). However, nowhere I could find asterisks.

Decision letter (RSOS-191356.R0)

06-Apr-2020

Dear Dr Jiang,

The editors assigned to your paper ("Effects of 9,10-phenanthrenequinone on antioxidant indices and metabolite profiles in *Takifugu obscurus* plasma") have now received comments from reviewers. We would like you to revise your paper in accordance with the referee and Associate Editor suggestions which can be found below (not including confidential reports to the Editor). Please note this decision does not guarantee eventual acceptance.

Please submit a copy of your revised paper before 29-Apr-2020. Please note that the revision deadline will expire at 00.00am on this date. If we do not hear from you within this time then it will be assumed that the paper has been withdrawn. In exceptional circumstances, extensions may be possible if agreed with the Editorial Office in advance. We do not allow multiple rounds of revision so we urge you to make every effort to fully address all of the comments at this stage. If deemed necessary by the Editors, your manuscript will be sent back to one or more of the original reviewers for assessment. If the original reviewers are not available, we may invite new reviewers.

Please also ensure that you upload your revised manuscript as an editable Word (.doc, .docx) or LaTeX file. Additionally, please ensure to upload all individual figure files in publication quality (PDF or .eps format).

If your study uses humans or animals please include details of the ethical approval received, including the name of the committee that granted approval. For human studies please also detail

whether informed consent was obtained. For field studies on animals please include details of all permissions, licences and/or approvals granted to carry out the fieldwork.

- Data accessibility

If you wish to submit your supporting data or code to Dryad (<http://datadryad.org/>), or modify your current submission to dryad, please use the following link:
<http://datadryad.org/submit?journalID=RSOS&manu=RSOS-191356>

- Competing interests

- Authors' contributions

- Acknowledgements

- Funding statement

Kind regards,
Lianne Parkhouse
Editorial Coordinator
Royal Society Open Science
openscience@royalsociety.org

on behalf of Dr Berat Haznedaroglu (Associate Editor) and Kevin Padian (Subject Editor)
 openscience@royalsociety.org

Subject Editor's comments:

Thanks for your submission. The reviewers had some useful comments and I hope you will attend to them specifically in your revised manuscript. In addition, the following points were raised:

- The justification for exposing the fish species to 50% of the 96 h LC50 is not provided or unclear. The authors should please supply the information.
- The basis for exposing the animal for a period of 96 h though being acute is not clear. Can non-target organisms be exposed to this level of oxygenated PAHs?

I hope you find these comments helpful, and best wishes for your revisions.

Reviewers' Comments to Author:

Reviewer: 1

Comments to the Author(s)

The research with title: Effects of 9,10-phenanthrenequinone on antioxidant indices and metabolite profiles in *Takifugu obscurus* plasma, was revised. The article focuses on studying the effect of 9,10-PQ in *T. obscurus*, as well as identifying the metabolic pathways that were influenced in response, allowing important information to be provided to clarify the effects of these pollutants on aquatic organisms. However, some points need to be improved and clarified in order to be able to consider publishing the work.

In the Introduction section:

1. On line 52, the authors mention the interest in assessing the effects of these pollutants on aquatic organisms. However, in reviewing references 7 and 8, it was found that these articles focus primarily on the effects on human populations, so it is advisable to look for other references focused on aquatic organisms.
2. Check line 53 to 55, the information presented does not match the original reference, since in this study they focused on studying *A. brevis*. In a part of the article they mention that the effects presented by this species is equivalent to the evaluation criteria of bioassays where *Rhepoxinius abronius* is used.

In the material and methods section:

1. On line 53 of page 2, justify the reason for using 50% of the LC50 of 9,10-PQ.
2. In line 55 to 56 on page 2 the authors mention that during the exhibition, the 75% of the culture media were renewed daily. The stability kinetic of 9,10-PQ was performed previously, what then is the actual concentration to which the organisms were exposed? Clarify this point.
3. Is the term "cultural media" correct? check the grammar
4. It is important that authors review the title of point 3.4 "Determination of antioxidant indices". Whereas MDA content levels are a biomarker of oxidative damage or index of damage. In contrast, enzyme activity (SOD and CAT) refers to enzymatic antioxidant status. Check the review the definition of antioxidant indices.

In the results section:

1. In section 4.1, it is recommended to indicate the percentages of the increase or decrease with

respect to the control in the times where a significant difference occurred, this in order to appreciate the magnitude of the change in the biomarkers of oxidative stress.

In the discussion section:

1. In line 50 on page 5, specify that the levels of LPO that increased in *P. vivipara* exposed to 200 ug/L phenanthrene were in the muscle.
2. Review the citation on line 59, page 5, [46] the paragraph does not correspond to that reference.
3. Review the numbering of the citations from page 6.

In the reference section:

1. Review reference 41, missing names of authors
2. Review the numbering of references, starting with reference 45

Reviewer: 2

Comments to the Author(s)

Please include the life stage of the fish species used. This is because a differential response might be observed across male and female or life stage.

Reviewer: 3

Comments to the Author(s)

Manuscript title: Effects of 9,10-phenanthrenequinone on antioxidant indices and metabolite profiles in *Takifugu obscurus* plasma

Manuscript ID: RSOS-191356

Takifugu obscurus is an interesting fish species due to its unique place in genomics and also due to its value in culinary industry. Authors have presented a comprehensive data on this species under exposure to 9,10-phenanthrenequinone. Manuscript is written in a good manner. However, there are few comments.

1. Selection on one PAH 9,10-phenanthrenequinone needs to be justified. Is this the dominant PAH? What about BaP that is benzo-a-pyrene?
2. Are plasma data sufficient? I mean, some vital tissue such as liver which is critical for metabolism should have been taken.
3. Some data show plateau such as MDA. Then how this response can be correlated with long-term exposure.
4. MDA (I guess lipid peroxidation) is described as an antioxidant (see 3.4 and 4.1).
5. In Figures 1,2 and 3 authors write that significant differences ($p < .05$) between control and 9,10-PQ treatment are marked by asterisks (Student's t-test). However, nowhere I could find asterisks.

Author's Response to Decision Letter for (RSOS-191356.R0)

See Appendix A.

RSOS-191356.R1 (Revision)

Review form: Reviewer 1

Is the manuscript scientifically sound in its present form?

Yes

Are the interpretations and conclusions justified by the results?

Yes

Is the language acceptable?

Yes

Do you have any ethical concerns with this paper?

No

Have you any concerns about statistical analyses in this paper?

No

Recommendation?

Accept as is

Comments to the Author(s)

When performing the second review of the manuscript, the author responded to the requested recommendations, so he considered the manuscript it's eligible for publication.

Review form: Reviewer 2

Is the manuscript scientifically sound in its present form?

Yes

Are the interpretations and conclusions justified by the results?

Yes

Is the language acceptable?

Yes

Do you have any ethical concerns with this paper?

No

Have you any concerns about statistical analyses in this paper?

No

Recommendation?

Accept as is

Comments to the Author(s)

Thank you for providing satisfactory responses to the review comments and updates in relevant sections of the manuscript.

Review form: Reviewer 3 (Sheikh Raisuddin)**Is the manuscript scientifically sound in its present form?**

Yes

Are the interpretations and conclusions justified by the results?

Yes

Is the language acceptable?

Yes

Do you have any ethical concerns with this paper?

No

Have you any concerns about statistical analyses in this paper?

No

Recommendation?

Accept as is

Comments to the Author(s)

RSOS-191356.R1

Effects of 9,10-phenanthrenequinone on antioxidant indices and metabolite profiles in *Takifugu obscurus* plasma

Authors have appropriately revised the manuscript and have also provided satisfactory responses to queries/comments.

Decision letter (RSOS-191356.R1)

Dear Dr Jiang,

It is a pleasure to accept your manuscript entitled "Effects of 9,10-phenanthrenequinone on antioxidant indices and metabolite profiles in *Takifugu obscurus* plasma" in its current form for publication in Royal Society Open Science. The comments of the reviewer(s) who reviewed your manuscript are included at the foot of this letter.

Please ensure that you send to the editorial office an editable version of your accepted manuscript, and individual files for each figure and table included in your manuscript. You can

send these in a zip folder if more convenient. Failure to provide these files may delay the processing of your proof. You may disregard this request if you have already provided these files to the editorial office.

on behalf of Dr Berat Haznedaroglu (Associate Editor) and Kevin Padian (Subject Editor)
openscience@royalsociety.org

Reviewer comments to Author:
Reviewer: 1

Comments to the Author(s)
When performing the second review of the manuscript, the author responded to the requested recommendations, so he considered the manuscript it's eligible for publication.

Reviewer: 3

Comments to the Author(s)
RSOS-191356.R1
Effects of 9,10-phenanthrenequinone on antioxidant indices and metabolite profiles in *Takifugu obscurus* plasma

Authors have appropriately revised the manuscript and have also provided satisfactory responses to queries/comments.

Reviewer: 2

Comments to the Author(s)
Thank you for providing satisfactory responses to the review comments and updates in relevant sections of the manuscript.

Appendix A

Subject Editor's comments:

Thanks for your submission. The reviewers had some useful comments and I hope you will attend to them specifically in your revised manuscript. In addition, the following points were raised:

- The justification for exposing the fish species to 50% of the 96 h LC₅₀ is not provided or unclear. The authors should please supply the information.

Re: The justification for exposing the fish species to 50% of the 96 h LC₅₀ is provided in manuscript named "Histological, oxidative and immune changes in response to 9,10-phenanthrenequinone, retene and phenanthrene in *Takifugu obscurus* liver" which was accepted by journal "Journal of Environmental Science and Health, Part A" but not published, and that was a more basic research. The justification in that manuscript was detailedly illustrated with data and figure. The exposure concentration was set at 50% 96 h-LC₅₀ as high exposure concentration to test the acute toxicity of 9,10-PQ, which could maximize the acute toxic effect as well as not cause death. We add this explanation in point

3.3 Exposure and sampling according to this comment.

- The basis for exposing the animal for a period of 96 h though being acute is not clear.

Can non-target organisms be exposed to this level of oxygenated PAHs?

Re: The duration of 96 h is a frequently-used period for acute toxicity test. In the

manuscript, we mentioned “Following the Gulf War oil spill event on 1991, PAHs could be detected in local seawater even after three years, with the concentration of total PAHs ranging from 21.14 µg/L to 320.5 µg/L [5]” in the introduction part, which meaning the sustainability of PAH compounds. This level of pollution could inevitably destroy the water ecological environment and influence diverse non-target organisms. In contrast, the duration of 96 h was a quite short period which could test the acute toxicity of contamination. Thus, we adopted it in our research.

I hope you find these comments helpful, and best wishes for your revisions.

Reviewers' Comments to Author:

Reviewer: 1

Comments to the Author(s)

The research with title: Effects of 9,10-phenanthrenequinone on antioxidant indices and metabolite profiles in *Takifugu obscurus* plasma, was revised. The article focuses on studying the effect of 9,10-PQ **on** *T. obscurus*, as well as identify the metabolic pathways was influenced in response, allowing important information to be provided to clarify the effects of these pollutants on aquatic organisms. However, some points need to be improved and clarified in order to be able to consider publishing the work.

In the Introduction section:

1. On line 52, the authors mention the interest in assessing the effects of these pollutants on aquatic organisms. However, in reviewing references 7 and 8, it was found that these articles focus primarily on the effects on human populations, so it is advisable to look for other references focused on aquatic organisms.

Re: In original manuscript, we considered the toxic effect of PAHs on human. Later we deleted this part for weak correlation, but forgot to delete the references. Reference 9 (Perspective on Ecotoxicology of PAHs to Fish) detailed the ecotoxicology of PAHs to Fish, which is enough to support the point of the sentence, hence we delete references 7 and 8.

2. Check line 53 to 55, the information presented does not match the original reference, since in this study they focused on studying *A. brevis*. In a part of the article they mention that the effects presented by this species is equivalent to the evaluation criteria of bioassays where *Rhepoxinius abronius* is used.

Re: We deleted the sentence from line 53 to 55 "For instance, acute exposure to benzo(a)pyrene, a typical PAH, induced significant mortality and chronic treatments with benzo(a)pyrene significantly inhibited the growth of *Rhepoxinius abronius*" and also deleted reference.

In the material and methods section:

1. On line 53 of page 2, justify the reason for using 50% of the LC₅₀ of 9,10-PQ.

Re: The justification for exposing the fish species to 50% of the 96 h LC₅₀ is provided in manuscript named "Histological, oxidative and immune changes in response to 9,10-phenanthrenequinone, retene and phenanthrene in *Takifugu obscurus* liver" which was accepted by journal "Journal of Environmental Science and Health, Part A" but not published, and that was a more basic research. The justification in that manuscript was detailedly illustrated with data and figure. The exposure concentration was set at 50% 96 h-LC₅₀ as high exposure concentration to test the acute toxicity of 9,10-PQ, which could maximize the acute toxic effect as well as not cause death. We add this explanation in point

3.3 Exposure and sampling according to this comment.

2. In line 55 to 56 on page 2 the authors mention that during the exhibition, the 75% of the culture media were renewed daily. The stability kinetic of 9,10-PQ was performed previously, what then is the actual concentration to which the organisms were exposed?

Clarify this point.

Re: In our experiment, we renewed the 75% of the culture media for two purposes. First, we had to keep the experimental water environment clean so as to avoid the interference of water environment pollution and ensure the reliability of experimental results. Besides, we had to avoid any death which may be caused by water pollution since the concentration

utilized in the experiment was high. Moreover, the replacement of the culture media was beneficial to maintain the exposure concentration, which could increase the accuracy of experimental results.

3. Is the term " cultural media" correct? check the grammar.

Re: We have replaced "cultural media" by "culture media" according this comment.

4. It is important that authors review the title of point 3.4 "Determination of antioxidant indices". Whereas MDA content levels are a biomarker of oxidative damage or index of damage. In contrast, enzyme activity (SOD and CAT) refers to enzymatic antioxidant status. Check the review the definition of antioxidant indices.

Re: We revised according to this comment and changed the title of point 3.4

"Determination of antioxidant indices" to "Determination of oxidative stress biomarkers".

Besides, we also changed the title of point 4.1 "Changes of antioxidant indices" to

"Changes of oxidative stress biomarkers".

In the results section:

1. In section 4.1, it is recommended to indicate the percentages of the increase or decrease with respect to the control in the times where a significant difference occurred, this in order to appreciate the magnitude of the change in the biomarkers of oxidative stress.

Re: We accepted this comment as added “at 72.97%, 91.21%, 85.84%, 93.37% and 97.44% respectively” after “MDA content”, added “with the increase of 45.18% and 44.54% respectively” after “at 6 h” and added “with the decrease of 29.86% and 32.62% respectively” after “at 96 hours” in section 4.1.

In the discussion section:

1. In line 50 on page 5, specify that the levels of LPO that increased in *P. vivipara* exposed to 200 ug/L phenanthrene were in the muscle.

Re: We revised this part to “Exposure to 200 µg/L phenanthrene significantly increased the LPO values in the muscle of estuarine guppy *Poecilia vivipara* at the level of Bayes Factor > 24” according to this comment.

2. Review the citation on line 59, page 5, [46] the paragraph does not correspond to that reference.

Re: There was a sequence error here, and “[46]” here should be replaced by “[45]”, and the order after “[45]” was also adjusted.

3. Review the numbering of the citations from page 6.

Re: The numbering of citations was adjusted after all the revisions were done.

In the reference section:

1. Review reference 41, missing names of authors

Re: We think the referee may mean reference 42 and the missing names of authors were added.

2. Review the numbering of references, starting with reference 45

Re: The numbering of references was adjusted after all the revisions were done.

Reviewer: 2

Comments to the Author(s)

Please include the life stage of the fish species used. This is because a differential response might be observed across male and female or life stage.

Re: We used juvenile *Takifugu obscurus* in our experiment. In this life stage, gonad is not mature to distinguish genders. "juvenile" was added in the first sentence of point 3.2 Animals according to this comment.

Reviewer: 3

Comments to the Author(s)

Manuscript title: Effects of 9,10-phenanthrenequinone on antioxidant indices and metabolite profiles in *Takifugu obscurus* plasma

Takifugu obscurus is an interesting fish species due to its unique place in genomics and also due to its value in culinary industry. Authors have presented a comprehensive data on this species under exposure to 9,10-phenanthrenequinone. Manuscript is written in a good manner. However, there are few comments.

1. Selection on one PAH 9,10-phenanthrenequinone needs to be justified. Is this the dominant PAH? What about BaP that is benzo-a-pyrene?

Re: Phenanthrene is one of the 16 PAHs which have been stipulated as the prior pollutants by the Environmental Protection Agency of USA and it is known as a dominant PAH. 9,10-phenanthrenequinone is an oxygenated derivate of phenanthrene (We have mentioned this in the first sentence of last paragraph in the 2. Introduction part.), thus it is an appropriate oxy-PAH for us to discuss.

2. Are plasma data sufficient? I mean, some vital tissue such as liver which is critical for metabolism should have been taken.

Re: We mentioned "Plasma is a pool of metabolites and can systematically reflect metabolic changes in the whole body [33]. Thus, plasma is a suitable subject to investigate physiological changes of organisms in response to pollutants" in our manuscript for the reason why we chose plasma. Some vital tissue could also provide useful metabolic information, but we want to systematically understand the metabolic

changes in the whole body, thus we thought plasma is the best and the most fundamental. In the future, we may carry out other research on the basis.

3. Some data show plateau such as MDA. Then how this response can be correlated with long-term exposure.

Re: In this acute toxicity test, MDA content of 9,10-PQ group increased rapidly after the start of the test, and plateaued from 12 h to 96 h, which mean the exposure concentration was high enough to trigger lipid peroxidation to the maximum extent rapidly. We are not sure if this steady-state of MDA content would continue after 96 h, but such a high exposure concentration would cause death as time goes on after 96 h based on the theory of toxicology.

4. MDA (I guess lipid peroxidation) is described as an antioxidant (see 3.4 and 4.1).

Re: We revised according to this comment and changed the title of point 3.4

“Determination of antioxidant indices” to “Determination of oxidative stress biomarkers”.

Besides, we also changed the title of point 4.1 “Changes of antioxidant indices” to “Changes of oxidative stress biomarkers”.

5. In Figures 1,2 and 3 authors write that significant differences ($p < .05$) between control and 9,10-PQ treatment are marked by asterisks (Student's t-test). However, nowhere I could find asterisks.

Re: This error may be due to the word app version. We also found that the manuscript we submitted did not include the asterisks, but in the attached pictures, the asterisks were existing. We have revised the pictures in the manuscript according to this comment.